# Effects of Bradykinin B2 Receptor Ablation from Tyrosine Hydroxylase Cells on Behavioral and Motor Aspects in Male and Female Mice

**DOI:** 10.3390/ijms25031490

**Published:** 2024-01-25

**Authors:** Thaina Maquedo Franco, Mariana R. Tavares, Leonardo S. Novaes, Carolina D. Munhoz, Jose Eduardo Peixoto-Santos, Ronaldo C. Araujo, Jose Donato, Michael Bader, Frederick Wasinski

**Affiliations:** 1Department of Neurology and Neurosurgery, Federal University of Sao Paulo, Sao Paulo 04039-032, Brazil; thaina.maquedo@hotmail.com (T.M.F.); mrtavares@unifesp.br (M.R.T.); peixoto.santos@unifesp.br (J.E.P.-S.); 2Department of Pharmacology, Instituto de Ciencias Biomedicas, Universidade de São Paulo, Sao Paulo 05508-000, Brazil; leonardo.s.novaes@gmail.com (L.S.N.); cdmunhoz@usp.br (C.D.M.); 3Department of Biophysics, Federal University of Sao Paulo, Sao Paulo 04039-032, Brazil; araujo.ronaldo@unifesp.br; 4Department of Physiology and Biophysics, Instituto de Ciencias Biomedicas, Universidade de São Paulo, Sao Paulo 05508-000, Brazil; jdonato@icb.usp.br; 5Max-Delbrück Center for Molecular Medicine (MDC), Robert-Rössle-Str. 10, 13125 Berlin, Germany; mbader@mdc-berlin.de; 6German Center for Cardiovascular Research (DZHK), Partner Site Berlin, 10117 Berlin, Germany; 7Institute for Biology, University of Lübeck, Ratzeburger Allee 160, 23562 Lübeck, Germany

**Keywords:** bradykinin, B2 receptor (B2R), tyrosine hydroxylase (TH) cells, metabolic aspects

## Abstract

The kallikrein–kinin system is a versatile regulatory network implicated in various biological processes encompassing inflammation, nociception, blood pressure control, and central nervous system functions. Its physiological impact is mediated through G-protein-coupled transmembrane receptors, specifically the B1 and B2 receptors. Dopamine, a key catecholamine neurotransmitter widely distributed in the CNS, plays a crucial role in diverse physiological functions including motricity, reward, anxiety, fear, feeding, sleep, and arousal. Notably, the potential physical interaction between bradykinin and dopaminergic receptors has been previously documented. In this study, we aimed to explore whether B2R modulation in catecholaminergic neurons influences the dopaminergic pathway, impacting behavioral, metabolic, and motor aspects in both male and female mice. B2R ablation in tyrosine hydroxylase cells reduced the body weight and lean mass without affecting body adiposity, substrate oxidation, locomotor activity, glucose tolerance, or insulin sensitivity in mice. Moreover, a B2R deficiency in TH cells did not alter anxiety levels, exercise performance, or motor coordination in female and male mice. The concentrations of monoamines and their metabolites in the substantia nigra and cortex region were not affected in knockout mice. In essence, B2R deletion in TH cells selectively influenced the body weight and composition, leaving the behavioral and motor aspects largely unaffected.

## 1. Introduction

Several biological processes, such as the inflammatory response [1], nociception, pain transmission [2], blood pressure control [3], cardiac hypertrophy, local blood flow, electrolyte and glucose transport, and cell proliferation [1] are modulated by the kallikrein–kinin system (KKS). Additionally, KKS is actively involved in gastrointestinal functions [4]. However, important for this work, this system regulates various brain functions [5].

Briefly, this KKS is composed of glycoprotein precursor substrates (kininogens); proteolytic enzymes from the serine protease family (tissue and plasma kallikrein), present in glandular cells, neutrophils, and biological fluids; and vasoactive peptides (kinins), derived from the hydrolysis of kininogens by kallikreins. The actions of tissue and plasma kallikreins in humans release the vasoactive peptides kallidin or Lys-BK (LBK) and bradykinin (BK). LBK is generated by the action of kallikreins on low-molecular-weight kininogen (LMWK), while BK is generated by the activity of kallikreins on a high-molecular-weight kininogen (HMWK) [6].

The physiological effects of kinins are mediated by two G-protein-coupled transmembrane receptors known as the B1 and B2 receptors (B1Rs and B2Rs, respectively). The B1R is normally poorly expressed but highly regulated in the presence of inflammatory stimuli or by des-Arg-kinins [7]. In contrast, the B2R is constitutively expressed under physiological conditions and is responsible for most of the effects of kinins [8]. The dysfunction of this receptor is implicated in cancer development and cardiovascular and neurological diseases [9]. Consequently, both the structure and pharmacology of B2Rs have become focal points in medical research [10]. Both kinin receptors are positively regulated under pathological conditions involved in proinflammatory effects [11]; for example, in temporal lobe epilepsy [12] or ischemic stroke [13]. Regarding the central nervous system (CNS), the expression of the B2R is well described in the hippocampal and olfactory bulb neurons of rats [14,15].

For almost five decades, researchers have provided evidence of the KKS’s involvement in pathological and physiological conditions in the brain [16]. The interest in the involvement of kinin B2Rs in pathophysiological conditions has evolved more rapidly than that of B1Rs, probably due to the B1R’s inflammatory profile [17]. Thus, it is considered that the B1R contributes to the pathogenesis of neurodegenerative diseases, while the B2R has been considered a neuroprotective factor [7,18]. Notably, the B2R plays a pivotal role in facilitating long-term neurogenesis [19,20], influencing the differentiation of neuronal cells, particularly in the hippocampus [21], and assuming a protective function against cell death [22].

Recently, our research group provided evidence that animals lacking the B1R and expressing only the B2R showed an increase in cell proliferation in the dentate gyrus after running, thus showing the importance of the B2R in mediating exercise-induced neuroprotective effects. Additionally, data also show that the absence of B2Rs alters exercise performance in mice [21]. Considering the pivotal role of the mesolimbic dopaminergic pathway in regulating locomotor activity in animal models [23,24], and given the fact that the B2R is able to physically interact with the D2 dopamine receptor (DrD2), with both being members of G protein-coupled receptors superfamily [25], we wonder if kinin signaling in the dopaminergic pathway could be one of the possible mechanisms involved in the altered locomotor activity of B2R knockout (KO) animals.

Dopamine is synthesized from tyrosine by tyrosine hydroxylase (TH) and then transported inside secretory vesicles for storage and release. A well-established model to study the catecholaminergic/dopaminergic pathway is through modulating gene expression from neurons expressing TH [26], the rate-limiting enzyme of catecholamine biosynthesis. The central aim of this study was to evaluate the effects of selective ablation of the B2R in catecholaminergic neurons, with a particular focus on neurons located in the substantia nigra (SN), given their crucial role in dopamine production and their significant abundance in this region. Investigating the effects of this ablation on the metabolic, behavioral, and motor aspects, we aim to provide a comprehensive overview of this animal model, contributing to the understanding of B2R implications in this process. Focusing our analysis on catecholaminergic neurons, especially those located in the SN, we can comprehensively clarify the role played by the B2R in this specific context.

## 2. Results

### 2.1. Generation of Mice Carrying Ablation of B2R in TH Neurons

To explore the consequences of B2R ablation in TH cells, control and TH^ΔB2R^ mice were generated. Considering the critical physiological importance of the mesolimbic dopaminergic pathway and the high number of TH neurons in the substantia nigra (SN), micropunches comprising the SN were obtained from 12-week-old mice. Initially, the B2R mRNA levels were quantified. We observed a significant reduction in B2R mRNA in the SN of TH^ΔB2R^ animals compared to the control group (Figure 1C). This result confirms the genetic ablation of the B2R in areas of the brain containing TH neurons. Importantly, B1R expression was not affected in TH^ΔB2R^ mice (Figure 1D), indicating the absence of compensatory adaptations of the KKS. The TH mRNA levels in the SN were similar between groups (Figure 1E). Since TH cells are also found outside in peripheral tissues, B2R mRNA was analyzed in the adrenal gland, and in this case, no significant reduction was observed compared to the control mice (Figure 1F). Together, these findings indicate a robust ablation of the B2R in major catecholaminergic neuronal populations without compensatory adaptations.

### 2.2. B2R Ablation in TH Cells Leads to a Transitory Reduction in Body Weight in Female and Male Mice

To verify the consequences of the B2R deletion in TH cells, the body weights and body compositions were assessed over time in both male and female mice. In the female group, a reduction in the body weight was observed at 5 and 10 weeks of age (Figure 2A). A reduction in the lean mass was noticed at 5 weeks of age (Figure 2B), while no change in fat mass was observed in the female mice (Figure 2C).

In the male group, we observed a reduction in the body weight (Figure 2E) and lean mass (Figure 2F) only in 5-week-old mice, while no change in the fat mass was observed (Figure 2G). No differences were observed at other ages compared to the control groups (Figure 2E–G). No changes in food consumption were verified in the TH^ΔB2R^ mice (Figure 2D,H). Thus, B2R ablation in TH cells impairs body growth in pubertal mice but does not cause major consequences in adult animals.

### 2.3. Impact of B2R Ablation in TH Cells on Metabolic Parameters

Either the KKS or the mesolimbic dopaminergic system are involved in the control of metabolism [27,28,29,30,31]. To investigate the potential impact of B2R deletion in cells expressing TH on energy homeostasis, we conducted an analysis of O_2_ consumption and CO_2_ production in both female and male mice. We did not observe significant changes in the metabolic parameters, including VO_2_, RER, and ambulatory activity in both female (Figure 3A–C) and male (Figure 3D–F) TH^ΔB2R^ mice. We also performed an area under the curve analysis for all parameters measured, and no change was found between the groups. Thus, B2R expression in TH neurons does not appear to be necessary for the maintenance of energy homeostasis.

### 2.4. Glucose Homeostasis Is Unaltered by B2R Ablation from TH Cells

KO mice for the B1R and B2R [32] as well as alterations in the dopaminergic neurons [28] result in modifications in glucose homeostasis. Therefore, glucose homeostasis was assessed in TH^ΔB2R^ mice. The Ablation of B2R in the TH cells did not affect the glucose tolerance test (Figure 4A,C) or insulin tolerance test, neither in the male or female mice (Figure 4B,D). 

### 2.5. B2R Ablation in TH Cells Does Not Affect Anxiety

Animal studies indicate that bradykinin may modulate anxiety [33,34,35]. Consequently, potential alterations in anxiety levels were scrutinized in TH^ΔB2R^ mice through the utilization of the open field and elevated plus maze tests (Figure 5). In the open field test, both female and male mice from the control and TH^ΔB2R^ groups exhibited a comparable frequency of entries into the center (Figure 5A,E). Additionally, no discernible disparity emerged in the duration spent in the center of the open field arena between the control and TH^ΔB2R^ mice (Figure 5B,F). Similarly, there were no significant distinctions in the number of entries and time spent in the open arms during the elevated plus maze test for both female (Figure 5C,D) and male (Figure 5G,H) mice. It is worth noting that the distance travelled and the duration of movement in the open field and elevated plus maze tests remained similar between control and TH^ΔB2R^ mice.

### 2.6. B2R Ablation in TH Cells Does Not Impair Physical Exercise Performance and Motor Coordination

In a B2R KO model, the absence of B2R appears to enhance performance in aerobic exercise [31]. Therefore, we subjected TH^ΔB2R^ mice to voluntary wheel tests, maximum effort tests on the treadmill, and motor coordination tests (rotarod; Figure 6). No differences were observed in the voluntary wheel tests for both female (Figure 6A) and male (Figure 6D) mice. Similarly, no differences were observed in the maximum effort tests on the treadmill for female (Figure 6B) and male (Figure 6E) mice, or in the motor coordination tests (Figure 6C,F).

### 2.7. Monoamine and Metabolite Concentrations Remain Unaltered Following B2R Ablation in TH Cells

The neuronal activity of monoamines is a pivotal indicator for various conditions encompassing depression, anxiety, and other neurodegenerative diseases that intricately influence the dopaminergic system [36]. In this study, we investigated the impact of B2R deletion in TH-expressing cells on the concentrations of monoamines and their metabolites in the SN and cerebral cortex of both female and male mice. Our findings revealed that B2R deletion in TH cells did not induce significant alterations in the concentrations of the monoamines and metabolites analyzed in the SN and cerebral cortex (Table 1).

## 3. Discussion

Kinins play a complex and important role in regulating several brain functions, including in pathological conditions. Niewiarowska-Sendo et al. (2017) described, in vitro, an intriguing physical interaction between the B2R and the DrD2, both members of the G protein-coupled receptor superfamily [25]. Although further studies are needed to fully understand this interaction in vivo, our study aimed to elucidate the impact of the B2R on the modulation of the dopaminergic pathway through its modulation in catecholaminergic neurons. Therefore, this investigation extended to assessing the subsequent impact on the metabolic, behavioral, and motor aspects in both male and female mice. To explore the potential functions of the B2R within this context, we used Cre–lox technology to selectively ablate the B2R from TH cells. Among other catecholamine-producing neurons, this approach targets the SN, a pivotal region for the dopaminergic neuron system, with crucial roles in various brain functions such as motor control, reward processing, and motivated behavior [37]. The B2R ablation in the SN was successfully confirmed by a reduction in B2R mRNA levels compared to the control mice. Thus, for the first time, we demonstrate the effects of the B2R on cells expressing TH in a specific deletion model. Importantly, B1R expression remained unchanged in TH^ΔB2R^ mice, discarding possible compensatory mechanisms. It is worth mentioning that although our main focus was to study the importance of B2Rs in dopaminergic neurons of the SN/VTA complex, B2R ablation may have occurred in any catecholaminergic neuron. We did not detect B2R deletion in the adrenal gland, possibly because of the absence of B2R expression in the TH cells of the adrenal medulla. However, we cannot rule out that the B2R was not deleted in other dopamine-, noradrenaline- and adrenaline-producing cells, because they all express TH. In this way, we assessed other aspects to offer a comprehensive understanding of the consequences of the B2R deletion in this novel model.

Surprisingly, upon deletion of the B2R in cells of the SN expressing TH, the TH^ΔB2R^ mice exhibited normal neurotransmitter levels in the brain. This suggests that the B2R does not play a direct role in dopamine synthesis, and consequently, the observed monoamine patterns in KO mice remained unaltered under physiologic conditions.

We noticed a decrease in the body weights and changes in the body compositions during the initial weeks of life in both male and female TH^ΔB2R^ mice, with no discernible differences in food consumption or energy metabolism. Existing evidence underscores the involvement of bradykinin in neurons governing the secretion of gonadotropin-releasing hormone (GnRH) [38]. Another neuropeptide implicated in stimulating GnRH release and thereby modulating reproduction and growth is kisspeptin, encoded by the Kiss1 gene [39]. Studies in mice have shown that part of the Kiss1 neurons are dopaminergic [40,41,42]. Consequently, we hypothesize that in our study, the deletion of the B2R in TH-expressing cells might have influenced the activity of Kiss1 neurons in the anteroventral periventricular nucleus/preoptic area (AVPV/PeN) region, where a substantial proportion of neurons coexpress TH, and consequently, modulate GnRH functions in growth and reproduction [42]. This speculation could potentially elucidate the observed decrease in the body weights of TH^ΔB2R^ mice. However, since the lower body weight in young TH^ΔB2R^ mice was transitory and not sustained throughout age, we did not extensively investigate this hypothesis. Further research would be essential to unravel the role of the B2R in Kiss1 neurons, paving the way for future avenues of exploration.

The involvement of the B2R in glucose homeostasis has been thoroughly examined through both in vivo and in vitro studies. Existing research, as summarized in the review by Gregnani et al. [43], establishes that B2R signaling enhances the glucose uptake in cells, affecting both adipose tissue and skeletal muscle. Several studies have indicated that changes in the glucose metabolism may affect dopaminergic projections in the midbrain, particularly in relation to obesity and insulin resistance [44,45]. However, here, we did not observe any alterations in the glucose metabolism and energy expenditure in TH^ΔB2R^ mice of both sexes. This suggests that B2R is not implicated, at least not in dopaminergic neurons and under normal conditions, in the modulation of energy metabolism, and indicates a peripheric role for the B2R in insulin secretion or sensitivity.

B2R is widely distributed in the CNS [14], and the dysfunction of this receptor is implicated in cancer development and cardiovascular and neurological diseases [9]. Consequently, both the structure and function of B2R have become focal points for future medicinal research [10]. Previous studies utilizing the Cre–lox technique have shown alterations independent of a Cre-mediated gene deletion [46,47,48,49]. In our study, the control group did not carry the Cre transgene, so we did not observe differences in motivation for exercise in animals expressing Cre, as all groups exhibited similar voluntary and forced activities.

The comprehension of how dopamine and its receptors modulate eating behaviors and physical activity has been clarified through investigations utilizing D2 receptor KO animal models. Generally, locomotor activity and anxiety-like behavior exhibit a positive correlation with dopamine levels [50]. Previous research has demonstrated that animals lacking D2 receptors, exposed to a high-fat diet and voluntary physical activity, manifested a substantial reduction in their activity levels and an increased susceptibility to obesity. This phenomenon underscores the pivotal role of D2 receptor inactivation in contributing to obesity through alterations in both energy expenditure and physical activity patterns [23]. In our investigation, we specifically examined cells expressing TH, the precursor of dopamine synthesis, with the ablation of the B2R under basal conditions. Our findings revealed no significant alterations in food consumption, and we observed no changes in voluntary and forced physical activities, or in motor coordination tests, across both female and male TH^ΔB2R^ mice groups.

Dopamine plays a significant role in the control of anxiety by influencing various neurochemical processes and brain circuits, and abnormal dopamine levels can heighten reactivity to stressors, exacerbating anxiety symptoms [51,52]. For example, pharmacological activation/inhibition of dopamine D1 receptor in VTA neurons of mice can alter anxiety-like behavior [53]. The intracerebroventricular (i.c.v) injection of bradykinin into the brains of rats has been shown to elevate anxiety-like behavior while diminishing social interaction [33]. A more expansive investigation conducted in both mice and humans revealed that bradykinin exerts a diverse range of effects on stress responses through B2R-mediated mechanisms, demonstrating antagonistic effects in mice and humans [35].

In our study, we did not detect differences in the outcomes of the open field and elevated plus maze tests, both widely employed for assessing anxiety-related behaviors. Traditionally, an increase in the time spent in the center of the open field and in the open arms, or an augmentation in the number of entries into these areas, are indicators of reduced anxiety-like behavior [54,55]. However, following the deletion of the B2R in cells expressing TH in the SN, the TH^ΔB2R^ mice exhibited anxiety levels comparable to those observed in the control group.

In summary, our study demonstrates that B2R ablation in cells expressing TH in both female and male mice resulted in transitory alterations in the body weights and body compositions. Importantly, these changes occurred without affecting the glucose metabolism and energy homeostasis. Additionally, there were no observed modifications in the voluntary physical activity, motor coordination, and anxiety levels, indicating a specific impact of B2R ablation in the catecholaminergic cells on body composition without significant systemic metabolic or behavioral alterations. One concern regarding our study is that we did not expose the control and TH^ΔB2R^ animals to metabolic/behavioral challenges such as a high-fat diet or specific stressors like restraint or food restriction. Future investigations are warranted to comprehensively assess the potential role of B2R signaling in TH-expressing neurons under these conditions.

While previous studies in vitro suggest interactions between kinins, especially the B2R and dopaminergic receptors, further investigations are warranted to fully unravel the physiological significance of kinin actions on TH neurons. Additionally, the involvement of the B1R in the observed results remains unclear and represents an avenue for further exploration. Further research considering these factors will contribute to a more comprehensive understanding of the complex regulatory mechanisms involving the B2R and B1R in TH-expressing neurons.

## 4. Materials and Methods

### 4.1. Mice

The present study used 12–14-week-old male and female mice. To induce the ablation of the B2R specifically in catecholaminergic neurons, mice carrying loxP-flanked B2R alleles [13] (here called Bdkrb2^flox/flox^ mice; The Jackson Laboratory, JAX stock #030446) were bred with TH^cre^ mice (stock #008601, JAX mice), [26]. TH^ΔB2R^ mice were homozygous for the loxP-flanked *B2R* alleles and carried one copy of the Cre transgene, leading to the ablation of the B2R only in catecholaminergic neurons. The control mice were littermates, homozygous for the loxP-flanked *B2R* alleles without carrying the Cre gene. All mice used in the experiments were genotyped using the tail tip collected during weaning.

### 4.2. Evaluation of Energy and Glucose Homeostasis

The body weights and body compositions were recorded in mice at 5, 10, and 15 weeks of age. The body composition was analyzed using time-domain nuclear magnetic resonance (LF50 body composition mice analyzer; Bruker, Ettlingen, Germany). Subsequently, mice were individually housed for acclimation. Then, the daily food intake was recorded. For the glucose homeostasis evaluation, mice were fasted for 4 h by removing food from the cage before each test. Basal glucose levels were determined (time 0), mice received i.p. injections of 2 g/kg of glucose or 1 IU/kg of insulin, followed by serial determinations of blood glucose levels using a glucose meter through samples collected from the tail tip.

### 4.3. Metabolic Parameters

Female and male mice were individually housed in the Comprehensive Lab Animal Monitoring System (Columbus Instruments, Columbus, OH, USA). After a 48 h acclimation period, O_2_ consumption (VO_2_), CO_2_ production (VCO_2_), respiratory exchange ratio (RER; CO_2_ production/O_2_ consumption), and ambulatory activity (by infrared sensors) were determined for 48 h.

### 4.4. Behavioral Tests

All behavioral tests were performed between 12:00 and 4:00 p.m., except for the voluntary wheel running, which was monitored daily. Experiments were conducted in a sound-proof, temperature-controlled room, with a dimerized light maintained at 65 LUX. After each experiment, the apparatuses were cleaned with 70% ethanol and air dried between each trial to minimize potential confounding odors or residues. Each behavioral test was conducted with a 7-day interval between them. The data obtained were recorded and analyzed using ANY-Maze 7.3 software (Stoelting Co., Wood Dale, IL, USA).

#### 4.4.1. Open Field (OFT) and Elevated Plus Maze Tests (EPM)

To assess anxiety patterns, we performed open field and elevated plus maze tests. For the open field test, mice were placed in an open field arena (40 cm [w] × 40 cm [d] × 30 cm [h]), and the number of entries and the time spent in the center of the arena were recorded for 5 min [56]. For the elevated plus maze test, mice were placed in the center of a maze that was elevated 40 cm from the ground and was composed of two opposite open arms measuring 35 cm × 4 cm, crossed by two closed arms of the same dimensions. The number of entries and the time spent in the open arms were measured for 5 min [56].

#### 4.4.2. Voluntary Wheel Running and Incremental Treadmill Maximal Running Tests

To evaluate the motor activity and aerobic capacity, we submitted animals to voluntary wheel running and incremental treadmill maximal running tests. For the voluntary wheel running test, mice were single housed in standard cages equipped with a running wheel. The running wheels were made available to the mice for voluntary exercise, and access to the running wheel was unlimited throughout the experiment. An acclimation period was provided to the mice to familiarize themselves with the running wheel environment. The running distance and time were monitored daily using digital counters for 7 consecutive days. The amount of running was recorded to assess the voluntary physical activity of the mice.

For the incremental treadmill maximal running test, mice were adapted on the treadmill for 3 consecutive days at 5 m/min speed for 10 min. The speed was chosen to avoid a training effect. After the adaptation period, mice were submitted to an incremental treadmill maximal running test. Starting at 10 m/min, the speed was increased by 3 m/min every 3 min. The test stopped when the animal was incapable of maintaining the running speed for more than 5 s. The time required to reach fatigue and the mean velocity of the test were recorded. From these data, we calculated the average aerobic capacities of the animals [48].

#### 4.4.3. Rotarod Test

The rotarod test assessed the animal’s motor coordination and balance (Insight Equipamentos Ltd., Ribeirão Preto, Brazil). Over a span of two days, mice were placed on a horizontal rod (30 cm [d]) that rotated around its long axis (rpm). On the second day, the duration that the mice spent walking forward on the rotarod until they fell was measured. Initially, the animals underwent 3 training attempts (each lasting 2 min) at 24 h intervals to establish the basal motor activity. Twenty-four hours after the last training attempt, the animals underwent 3 motor test attempts, with a maximum duration of 5 min each and a 30 min interval between attempts. The rotation speed increased in 8 stages (4–38 rpm). The average latency to fall was recorded, and the animals that remained on the rotarod until the maximum time of each attempt received a score of 120 (for training) or 300 (for the test) seconds. The fall latency observed with the rotational cylinder provides valuable insights into the animal’s motor behavior, with a shorter fall latency typically indicating a decrease in motor coordination [57].

### 4.5. HPLC-Electrochemical Detection and Sample Preparation

The content of monoamines was identified and quantified using an HPLC system with electrochemical detectors previously described by Castro Neto et al. [58]. The substantia nigra (SN) and cortex region were dissected bilaterally, placed on an ice-chilled plate, weighed, and stored at −80 °C until assay. Tissues were ultrasonically homogenized in a 0.1 mol/L solution of HClO4 containing 0.02% Na_2_S_2_O_2_ (15 µL of solution for each milligram of tissue) and dihydroxybenzyl-amine (DHBA, 146.5 ng/mL) as the internal standard for monoamines. The samples were centrifuged at 11,000× *g* at 4 °C for 40 min, and then the supernatant was filtered and injected into an HPLC system. The monoamines NA, DA, and 5-HT, as well as their metabolites, were quantified as previously described [58]. Briefly, a Shimadzu LC-10AD isocratic system was employed, with a 20 µL injection loop and a Spheri-5 RP-18 5 µm column (220 × 4.6 mm), using electrochemical detection at 0.75 V and a mobile phase composed of phosphate/citrate (pH 2.64, 0.02 mol/L), 0.12 mmol/L ethylene diamine tetraacetic acid, and 0.06% heptane sulphonic acid, in 10% methanol, at a flow rate of 1 mL/min. The concentrations of monoamines and metabolites were expressed as the mean ± SEM ng/mg of wet tissue. Standard concentrations of monoamines and metabolites were tested throughout the runs, and the retention time was verified for each substance to certify that there were no peaks overlapping on sample delivery and were expressed as the mean ± SEM (nmol/L per milligram) of wet tissue.

### 4.6. qRT-PCR

For each experimental group (*n* = 6 per group), designated regions from the whole SN were dissected and pooled. Total RNA was isolated from the samples using TRIzol (Invitrogen, Waltham, MA, USA). The quantity and quality of the RNA were assessed using an Epoch microplate spectrophotometer (Biotek, Santa Clara, CA, USA). To remove any potential genomic DNA contamination, the total RNA was incubated with DNase I RNase-free (Roche Applied Science, Penzberg, Germany). For reverse transcription, 2 μg of total RNA was used with SuperScript II Reverse Transcriptase (Invitrogen) and random primers p(dN)6 (Roche Applied Science). Real-time PCR was performed on a 7500TM real-time PCR system (Applied Biosystems, Waltham, MA, USA) using Power SYBR Green (Applied Biosystems) as the fluorescent dye for detection. Relative quantification of mRNA expression was achieved using the 2^−∆∆Ct^ method, with *Actb* (β-actin) and *Ppia* (cyclophilin A) as the reference genes. The following primer sets were used for amplification:

*Actb* (forward: gctccggcatgtgcaaag; reverse: catcacaccctggtgccta), *Ppia* (forward: cttcttgctggtcttgccattcc; reverse: tatctgcactgccaagactgagt), *Bdkrb2* (forward: cctttctgggccatcaccat; reverse: ggtagcggtcgatactcacg), *Bdkrb1* (forward: tggagttgaacgttttgggttt; reverse: gtgaggatcagccccatt), and *Th* (forward: tccaatacaagcagggtgagc; reverse: ggcaggcatgggtagcatag). This qRT-PCR protocol was used to assess the expression levels of specific genes in the SN, which is a crucial brain region involved in the dopaminergic system.

### 4.7. Statistical Analysis

For comparisons between the two groups, the unpaired, two-tailed Student’s *t*-test was used. Changes in the glucose and insulin tolerance tests and metabolic parameters were analyzed using repeated measures two-way ANOVA. A post hoc analysis was performed using Bonferroni’s multiple comparisons test to assess the significance of differences between specific groups. The statistical analyses were carried out using GraphPad Prism 8.0 software. A *p*-value of less than 0.05 was considered statistically significant. The results are presented as the mean ± SEM (standard error of the mean), which provides a measure of the precision of the mean estimate.

## Figures and Tables

**Figure 1 ijms-25-01490-f001:**
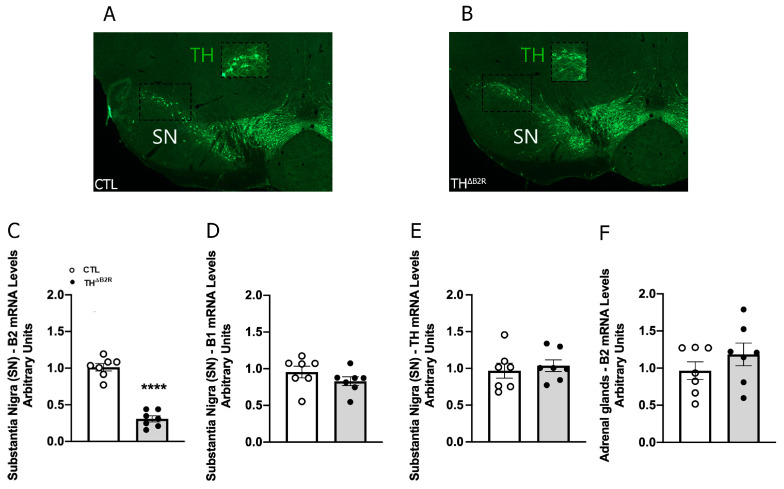
Validation of the TH^ΔB2R^ mouse model: B2R mRNA ablation in TH cells in the substantia nigra (SN). Localization of TH neurons in the substantia nigra (SN) and gene expression in control and TH^ΔB2R^ animals. Photomicrographs indicating the locations of TH neurons (green) in control group (**A**) and TH^ΔB2R^ group (**B**); TH immunoreactivity has been previously described by [26]. Bar graphs comparing the mRNA expression of the B2R (**C**), B1R (**D**), and TH (**E**) in the SN region. mRNA expression of the B2R in the adrenal gland (**F**) (control, *n* = 7, TH^ΔB2R^, *n* = 7). Data expressed as the mean ± SEM. **** *p* < 0.0001.

**Figure 2 ijms-25-01490-f002:**
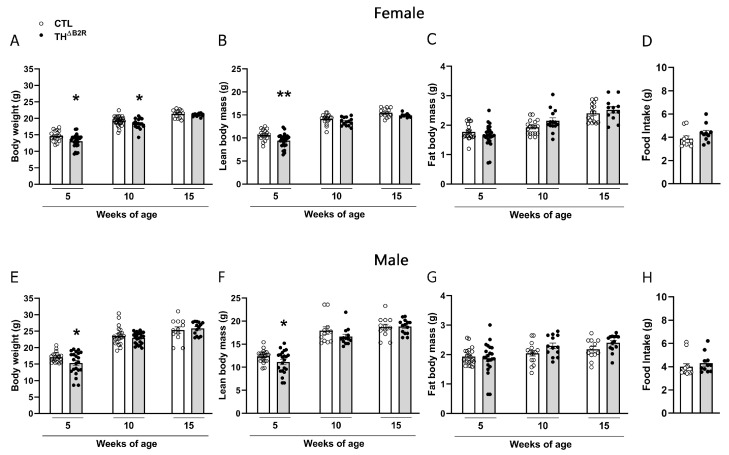
Bradykinin B2 receptor (B2R) ablation from TH cells (TH^ΔB2R^) altered the body weights of female and male mice. Body weight (**A**,**E**); lean body mass (**B**,**F**); and fat body mass (**C**,**G**) at 5, 10, and 15 weeks of age in control (*n* = 17–22) and TH^ΔB2R^ (*n* = 28–32) groups. Food consumption of female (**D**; *n* = 10) and male (**H**; *n* = 11–13) mice. Data expressed as the mean ± SEM. * *p* < 0.05; ** *p* < 0.01.

**Figure 3 ijms-25-01490-f003:**
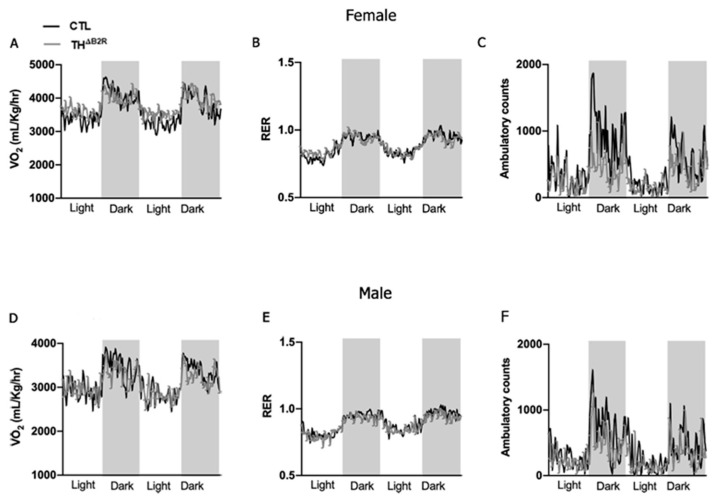
Bradykinin B2 receptor (B2R) ablation from TH cells (TH^ΔB2R^) is not necessary for the maintenance of metabolic parameters in mice. Oxygen consumption (VO_2_; **A**); respiratory exchange ratio (RER; **B**), and ambulatory activity (**C**) in control (*n* = 7) and TH^ΔB2R^ (*n* = 9) female mice. Oxygen consumption (VO_2_; **D**); respiratory exchange ratio (RER; **E**), and ambulatory activity (**F**) in control (*n* = 8) and TH^ΔB2R^ (*n* = 10) male mice. Differences between groups were analyzed using repeated measures two-way ANOVA.

**Figure 4 ijms-25-01490-f004:**
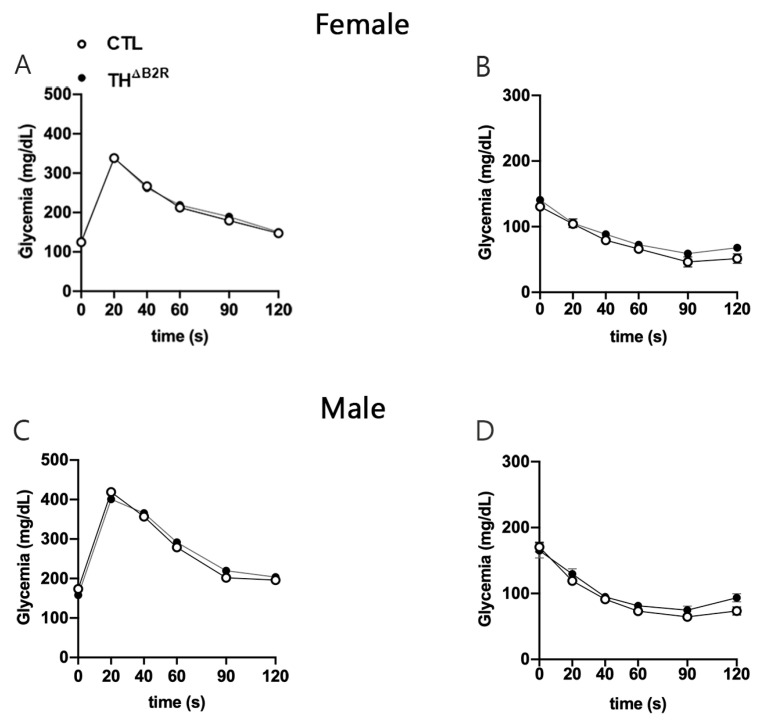
Glucose homeostasis is unaltered by bradykinin B2 receptor (B2R) ablation from TH cells (TH^ΔB2R^) in female and male mice. Glucose tolerance test (GTT; **A**) and insulin tolerance test (ITT; **B**) in control (*n* = 9) and TH^ΔB2R^ (*n* = 10) female mice. Glucose tolerance test (GTT; **C**) and insulin tolerance test (ITT; **D**) in control (*n* = 12) and TH^ΔB2R^ (*n* = 9) male mice. Possible differences between groups were analyzed using repeated measures two-way ANOVA.

**Figure 5 ijms-25-01490-f005:**
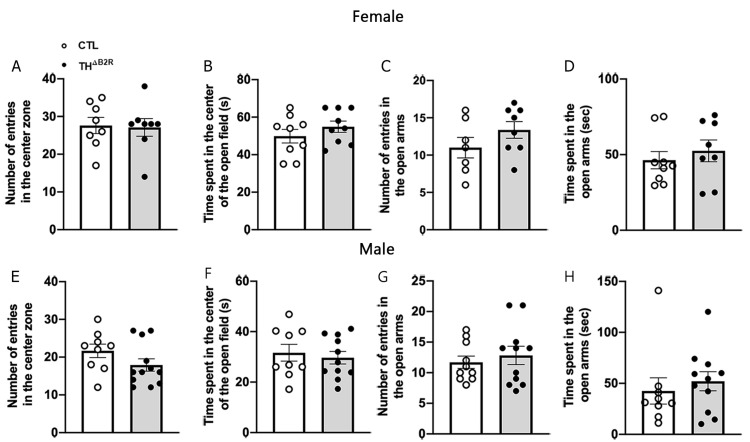
Bradykinin B2 receptor (B2R) ablation from TH cells does not modulate anxiety. Number of entries (**A**) and time spent (**B**) in the center of the open field in control (*n* = 7–9) and TH^ΔB2R^ (*n* = 7–9) female mice. Number of entries (**C**) and time spent (**D**) in the open arms during the elevated plus maze test in the control (*n* = 7–9) and TH^ΔB2R^ (*n* = 7–9) female mice. Number of entries (**E**) and time spent (**F**) in the center of the open field in the control (*n* = 9–10) and TH^ΔB2R^ (*n* = 11–12) male mice. Number of entries (**G**) and time spent (**H**) in the open arms during the elevated plus maze test in the control *(n* = 9–10) and TH^ΔB2R^ (*n* = 11–12) male mice. Differences between groups were analyzed using an unpaired, two-tailed Student’s *t*-test.

**Figure 6 ijms-25-01490-f006:**
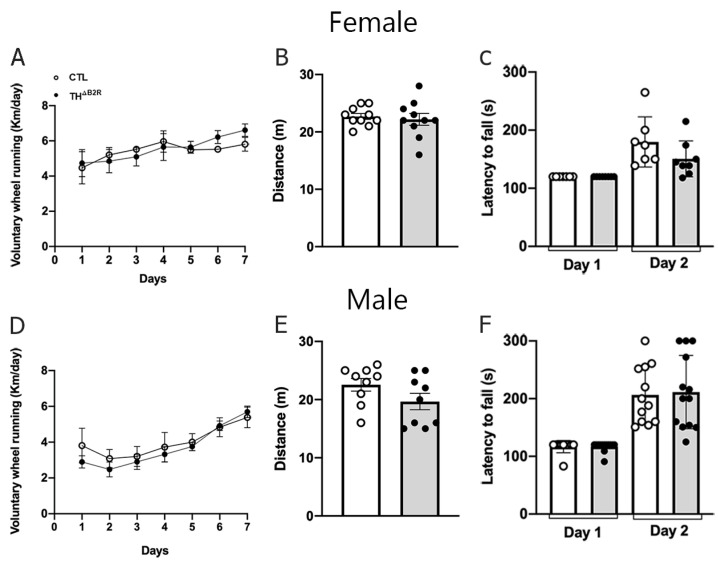
Effect of bradykinin B2 receptor (B2R) ablation from TH cells (TH^ΔB2R^) in mice on exercise performance and motor coordination. Running distance in voluntary wheels (**A**), incremental treadmill maximal running (**B**), and retention time on the rod in the rotarod test (**C**) in control (*n* = 6–10) and TH^ΔB2R^ (*n* = 6–11) female mice. Running distance in voluntary wheels (**D**), incremental treadmill maximal running (**E**), and retention time on the rod in the rotarod test (**F**) in control (*n* = 8–10) and TH^ΔB2R^ (*n* = 8–11) male mice. Differences between groups were analyzed using an unpaired, two-tailed Student’s *t*-test.

**Table 1 ijms-25-01490-t001:** Concentrations of monoamines and their metabolites in the substantia nigra (SN) and cortex region of TH^ΔB2R^ and control animals. The concentrations are expressed as ng/g of wet tissue. Substantia nigra (SN); 4-hydroxy-3-methoxy mandelic acid (VMA); norepinephrine (NA); (3,4-dihydroxyphenyl)-L-alanine (L-Dopa); 3,4-dihydroxyphenylacetic acid (DOPAC); dopamine (DA); 4-hydroxy-3-methoxy-phenylacetic acid (HVA); 5-hydroxyindoleacetic acid (5HIAA); 5-hydroxytryptamine (5HT). Student’s *t* test was performed to compare the control (*n* = 10) and TH^ΔB2R^ (*n* = 12) groups. Data are presented as the mean ± SEM.

		VMA	NA	LDOPA	DOPAC	DA	HVA	5HIAA	5HT
SN	Control	0.795 ± 0.53	0.660 ± 0.24	0.020 ± 0.06	0.235 ± 0.14	0.170 ± 0.04	0.245 ± 0.16	2.815 ± 0.85	0.800 ± 0.30
	TH^ΔB2R^	0.555 ± 0.33	0.580 ± 0.30	0.035 ± 0.33	0.180 ± 0.84	0.125 ± 0.55	0.240 ± 0.36	2.840 ± 0.56	0.710 ± 0.20
	P (T test)	0.368	0.933	0.455	0.434	0.345	0.503	0.504	0.780
Cortex	Control	0.560 ± 0.07	0.780 ± 0.10	0.020 ± 0.10	0.080 ± 0.11	0.05 ± 0.08	0.190 ± 0.10	1.430 ± 0.65	0.530 ± 0.10
	TH^ΔB2R^	0.655 ± 0.20	0.625 ± 0.35	0.015 ± 0.03	0.060 ± 0.04	0.030 ± 0.02	0.125 ± 0.03	1.305 ± 0.67	0.420 ± 0.18
	P (T test)	0.385	0.939	0.218	0.156	0.145	0.066	0.570	0.163

## Data Availability

The data supporting the findings of this study are available from the corresponding author upon reasonable request.

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
