# Peer review of "Effects of Bradykinin B2 Receptor Ablation from Tyrosine Hydroxylase Cells on Behavioral and Motor Aspects in Male and Female Mice"

_ijms, 2024, doi:10.3390/ijms25031490_

Round 1
Reviewer 1 Report
Comments and Suggestions for Authors
The manuscript from Maquedo Franco and Cols describes the effects of the selective ablactation of TH on body weight, oxygen consumption, glucose, anxiety, motor behavior, and monoamine metabolites in a mouse model. Despite that, except for a light transient body growth decrement in puberty, the rest of the determined parameters remain unchanged; the manuscript has several concerns.
1. The ablation was conducted on all catecholaminergic neurons. Thus, possible changes are attributed to nigral neurons only. This is incorrect.
2. The nigral neurons, to actual knowledge, did not participate in oxygen metabolism
3. What is the rationale for associating changes in the determined parameters and the nigral neurons?
4, The discussion is unarticulated and speculative and does not explain the results clearly.
5. We felt that a manuscript, first of the characterization of the mice model and the catecholaminergic neurons affected, and of the behavioral and biochemical consequences of the ablation, should be done in order to try to understand the role of the receptor in the nigral or all the catecholaminergic neurons.
6. the contribution of the knowledge of the receptor in nigral neurons is poor.
Author Response
RESPONSE TO REVIEWER 1
Reviewer #1: ( Comments and Suggestions for Authors)
The manuscript from Maquedo Franco and Cols describes the effects of the selective ablactation of TH on body weight, oxygen consumption, glucose, anxiety, motor behavior, and monoamine metabolites in a mouse model. Despite that, except for a light transient body growth decrement in puberty, the rest of the determined parameters remain unchanged; the manuscript has several concerns.
We would like to thank the reviewer for the time spent evaluating our manuscript and his/her comments that helped to improve the manuscript.
Specific comments:
Major:
- The ablation was conducted on all catecholaminergic neurons. Thus, possible changes are attributed to nigral neurons only. This is incorrect.
RESPONSE: Thank you for this observation. The reviewer is correct in saying that our genetic ablation occurred in all catecholaminergic neurons and not in specific groups of dopaminergic neurons. This information was clearly stated in the last paragraph of the introduction when we indicated our objective:
Line 85: “The objective of the present study was to investigate whether B2R ablation in catecholaminergic neurons affects metabolic, behavioral and motor aspects in male and female mice”
The ablation of B2R was assessed not only in the nigral neurons but also in the adrenal gland, which also contains many TH-expressing cells. In this case, no B2R deletion was detected (Fig. 1F). In addition, added a phrase in the Discussion Section stating that although our main focus was to study the nigral neurons and their physiological functions, other catecholaminergic neurons were likely affected, including TH neurons of the hypothalamus that co-express the Kiss1 gene, as discussed in our paper (third paragraph of the discussion, line 245).
Line 230: “It is worth mentioning that although our main focus was to study the importance of B2R in dopaminergic neurons of the SN/VTA complex, B2R ablation may have occurred in any catecholaminergic neuron. We did not detect B2R deletion in the adrenal gland, possibly because of the absence of B2R expression in TH cells of the adrenal medulla. However, we cannot rule out that B2R was not deleted in other dopamine-, noradrenaline- and adrenaline-producing cells because they all express TH”.
- The nigral neurons, to actual knowledge, did not participate in oxygen metabolism
RESPONSE: Certainly, the substantia nigra is not directly implicated in oxygen metabolism, and we did not claim otherwise in the article. The reason for investigating metabolism parameters in the animals is linked to the observation of reduced body weight in the early weeks of age following the deletion of the B2 receptor in the substantia nigra. We believed that this observation might be associated with potential metabolic implications arising from the genetic manipulation in the substantia nigra. Since the goal of the study was to assess metabolic, behavioral, and motor parameters under basal conditions with the deletion of the B2 kinin receptor in the substantia nigra, the evaluation of oxidative metabolism was performed.
- What is the rationale for associating changes in the determined parameters and the nigral neurons?
RESPONSE: Studies have demonstrated the interaction between the B2 kinin receptor and dopamine D2 receptors in vitro (Niewiarowska-Sendo et al., Biochim Biophys Acta Mol Cell Res. 2017;1864:1855-66). Given the extensive research on the catecholaminergic pathway, we chose to initiate our investigations using the TH/cre model. This model has been extensively studied, offering an effective way to deepen our comprehension of outcomes in both in vitro and in vivo settings. Consequently, we conducted a comprehensive characterization of this model, addressing metabolic, behavioral, and motor aspects to gain a more complete understanding of its implications.
- The discussion is unarticulated and speculative and does not explain the results clearly.
RESPONSE: We sincerely thank you for your detailed review and insightful comments on the discussion section of our work. Your feedback is invaluable to us, and we have carefully revised the discussion to enhance clarity and provide a clearer understanding of the results.
- We felt that a manuscript, first of the characterization of the mice model and the catecholaminergic neurons affected, and of the behavioral and biochemical consequences of the ablation, should be done in order to try to understand the role of the receptor in the nigral or all the catecholaminergic neurons.
RESPONSE: This is the first model we are exploring in our laboratory. Considering the earlier data that suggest B2R could play a role in functions regulated by the dopaminergic pathway and the possible in vivo interaction between B2R and D2R, we decided to investigate the deletion of this receptor in cells expressing TH. So far, our data indicate that, at least in this region of the substantia nigra, particularly in catecholaminergic neurons, we have not observed significant differences in motor and behavioral patterns. It's important to emphasize that this lack of changes does not invalidate the studied model; on the flip side, it contributes to the clarity of the role of the B2 receptor in the substantia nigra. This is an initial step in a series of models we are mapping, and the information obtained so far is crucial for guiding our and others future investigations. Studying the role of the B2 kinin receptor in catecholaminergic neurons not only deepens our understanding of neural mechanisms but also holds significant implications for mental health and the potential development of therapeutic strategies.
- the contribution of the knowledge of the receptor in nigral neurons is poor.
RESPONSE: This study represents a significant portion of our comprehensive effort to map the B2 kinin receptor in the dopaminergic pathway. By focusing on the substantia nigra, where there is significant expression of both TH and B2 kinin receptor cells, we have demonstrated that the deletion of B2R does not induce alterations in this specific context. Nevertheless, we are currently conducting additional research to deepen our understanding of this intricate relationship. These further investigations aim to provide a more complete and detailed insight into the role of the B2 kinin receptor in the dopaminergic pathway, thereby solidifying our comprehension of its influence in various regions of the brain.

Reviewer 2 Report
Comments and Suggestions for Authors
This research group previously reported that animals lacking the B1R and expressing only the B2R showed an increase in cell proliferation in the dentate gyrus after running, and that the absence of B2R alters exercise performance. B2R as a G-protein-coupled receptor could be physically interact with dopamine D2 receptor that regulates movement and behaviors in dopamine neurotransmission. Based on the previous findings, this study aims to explore whether mice with B2R KO TH+ cells (catecholamine cells) show changes in locomotor activity, metabolism, and behavior. The results showed that the mice with B2R ablation in TH+ cells revealed only reduced body weight and lean mass without affecting locomotor activity, glucose torelance, insulin sensitivity, and exercise performance. The results are of graat interst. Howevr, the following points should be considered in this article.
1) The results show only the significant reductions in body weight and lean mass, but in very modest degrees. The physiological significance is questioned. Althogh enough number of mice is used, the reproducibily of these data should be reconfirmed.
2) As authors stated, the weakness of this experimnet is without food restriction. The amounts of food consumed are not measured. The confirmation experimet should be done with food restriction.
3) The mechanism of reductions only in body weight and in lean mass are interesting in clinical points of view to develop anti-obecity drugs. Although the mechanism be explored in the next experiments, the authors could describe the possible mechanism in termes of B2R and dopamine D2R.
Comments on the Quality of English Language
Minor typographical errors should be examined.
Author Response
Reviewer #2 : ( Comments and Suggestions for Authors)
This research group previously reported that animals lacking the B1R and expressing only the B2R showed an increase in cell proliferation in the dentate gyrus after running, and that the absence of B2R alters exercise performance. B2R as a G-protein-coupled receptor could be physically interact with dopamine D2 receptor that regulates movement and behaviors in dopamine neurotransmission. Based on the previous findings, this study aims to explore whether mice with B2R KO TH+ cells (catecholamine cells) show changes in locomotor activity, metabolism, and behavior. The results showed that the mice with B2R ablation in TH+ cells revealed only reduced body weight and lean mass without affecting locomotor activity, glucose torelance, insulin sensitivity, and exercise performance. The results are of graat interst. Howevr, the following points should be considered in this article.
We would like to thank the reviewer for the time spent evaluating our manuscript and for their constructive comments and suggestions.
- The results show only the significant reductions in body weight and lean mass, but in very modest degrees. The physiological significance is questioned. Althogh enough number of mice is used, the reproducibily of these data should be reconfirmed.
RESPONSE: The results shown include multiple cohorts of animals, which explains the large sample sizes of the experimental groups (n = 17-32/group). Thus, the difference between control and THΔB2R mice was extensively confirmed in different experiments. Furthermore, similar findings were observed in male and female mice, which further supports the phenotype observed.
- As authors stated, the weakness of this experimnet is without food restriction. The amounts of food consumed are not measured. The confirmation experimet should be done with food restriction
RESPONSE: Food intake was assessed in our study, and no differences were observed between the experimental groups (Fig. 2D,H). In the manuscript, we acknowledged that we did not challenge the mice in conditions that disturb their homeostasis or are related to stress. Food restriction is an example of challenge that we mentioned. However, our primary focus in the current study was to evaluate the effects of B2 receptor deletion under baseline (non-stressed) conditions. We intend to address to investigate different situations of stress in future studies but it is out of the scope of the present work.
Line 309: “One concern regarding our study is that we did not expose these animals to metabolic/behavioral challenges such as a high-fat diet or specific stressors like restraint or food restriction. Future investigations are warranted to comprehensively assess the potential role of B2R signaling in TH-expressing neurons under these conditions”
- The mechanism of reductions only in body weight and in lean mass are interesting in clinical points of view to develop anti-obecity drugs. Although the mechanism be explored in the next experiments, the authors could describe the possible mechanism in termes of B2R and dopamine D2R.
RESPONSE: While in vitro studies have already demonstrated the physical interaction between the B2 kinin receptor and the dopamine D2 receptor, confirming the B2R–D2R dimerization could open new perspectives in modulating various cellular functions dependent on their activation. Our laboratory aims to gain a deeper understanding of this interaction using an in vivo model (cre/lox technology) to comprehend better the role of kinins in the central nervous system. Furthermore, considering that alterations in body weight and lean mass were temporary during a particular life phase (puberty), we thought it would be premature to elucidate potential mechanisms. This is based on the current literature regarding how B2 kinin receptors might influence the dopaminergic pathway through their interaction with the dopamine D2 receptor. To provide further clarity, we have made a modification in the following paragraph:
Line 216: “Kinins play a complex and important role in regulating several brain functions, including in pathological conditions. Niewiarowska-Sendo et al. (2017) described in vitro an intriguing physical interaction between the B2R and the DrD2, both members of the G protein-coupled receptors superfamily [25]. Although further studies are needed to fully understand this interaction in vivo, our study aimed to elucidate the impact of the B2R on the modulation of the dopaminergic pathway. This investigation extended to assessing the subsequent impact on metabolic, behavioral and motor aspects in both male and female mice. To explore this hypothesis, we used Cre-lox technology to selectively ablate B2R from TH cells. This approach targets the SN, a pivotal region for the dopaminergic neuron system, with crucial roles in various brain functions such as motor control, reward processing, and motivated behavior [37]. The B2R ablation in the SN was successfully confirmed by a reduction in B2R mRNA levels compared to control mice”.

Round 2
Reviewer 1 Report
Comments and Suggestions for Authors
The authors did not provide enough argumentation to my concerns. Although the animal model is interesting, the focus of the manuscript is still incomprehensible.
Author Response
Reviewer #1: ( Comments and Suggestions for Authors)
The authors did not provide enough argumentation to my concerns. Although the animal model is interesting, the focus of the manuscript is still incomprehensible.
RESPONSE: Thank you for your continued engagement with our manuscript, we appreciate the opportunity to address your concerns. We have carefully considered your feedback, and we understand your perspective regarding the focus and argumentation within the manuscript. We would like to provide further clarification to ensure a better understanding of our work.
We acknowledge your concern about the comprehensibility of the manuscript's focus. Our primary objective was to investigate the effects of selective ablation of tyrosine hydroxylase (TH) on various physiological and behavioral parameters, focusing on the substantia nigra's B2 kinin receptor. However, we understand that the emphasis on the substantia nigra may have led to confusion.
To address this concern, we have revised the manuscript to explicitly state our main research question, objectives, and the significance of studying the B2 kinin receptor in the dopaminergic pathway. We hope that these revisions will enhance the overall clarity and focus of our work.
Line 84: “The central aim of this study was to evaluate the effects of selective ablation of the B2R in catecholaminergic neurons, with a particular focus on neurons located in the substantia nigra (SN), given their crucial role in dopamine production and their significant abundance in this region. Investigating the effects of this ablation on metabolic, behavioral, and motor aspects, we aim to provide a comprehensive overview of this animal model, contributing to the understanding of B2R implications in this process. Focusing our analysis on catecholaminergic neurons, especially those located in the SN, we can comprehensively clarify the role played by B2R in this specific context.”
We appreciate your feedback on the need for more argumentation to address your concerns. In response, we have revisited the discussion section, providing additional rationale and context for our experimental design, including the choice of the TH/cre model and the specific focus on the substantia nigra.
We have expanded on the scientific literature supporting our investigation, particularly the existing knowledge of the interaction between the B2 kinin receptor and dopamine D2 receptors. By strengthening the argumentation, we aim to provide a more robust foundation for our research approach.
We hope that these clarifications and revisions will address your concerns and improve the overall comprehensibility of the manuscript. Your feedback is invaluable to us, and we are committed to ensuring that our work meets the highest standards of scientific rigor and clarity.
Line 221: “Kinins play a complex and important role in regulating several brain functions, including in pathological conditions. Niewiarowska-Sendo et al. (2017) described in vitro an intriguing physical interaction between the B2R and the DrD2, both members of the G protein-coupled receptors superfamily [25]. Although further studies are needed to fully understand this interaction in vivo, our study aimed to elucidate the impact of the B2R on the modulation of the dopaminergic pathway through its modulation in catecholaminergic neurons. Therefore, this investigation extended to assessing the subsequent impact on metabolic, behavioral and motor aspects in both male and female mice. To explore potential functions of B2R within this context, we used Cre-lox technology to selectively ablate B2R from TH cells. Among other catecholamine-producing neurons, this approach targets the SN, a pivotal region for the dopaminergic neuron system, with crucial roles in various brain functions such as motor control, reward processing, and motivated behavior [37]. The B2R ablation in the SN was successfully confirmed by a reduction in B2R mRNA levels compared to control mice. Thus, for the first time, we demonstrate the effects of the B2R on cells expressing TH in a specific deletion model. Importantly, B1R expression remained unchanged in THΔB2R mice, discarding possible compensatory mechanisms. It is worth mentioning that although our main focus was to study the importance of B2R in dopaminergic neurons of the SN/VTA complex, B2R ablation may have occurred in any catecholaminergic neuron. We did not detect B2R deletion in the adrenal gland, possibly because of the absence of B2R expression in TH cells of the adrenal medulla. However, we cannot rule out that B2R was not deleted in other dopamine-, noradrenaline- and adrenaline-producing cells because they all express TH. In this way, we assessed other aspects to offer a comprehensive understanding of the consequences of the B2R deletion in this novel model.”
Thank you once again for your time and thoughtful consideration.

Reviewer 2 Report
Comments and Suggestions for Authors
The authors revised the article as much as posible in adding the discussion by following the commnents of reviewers. Although this article is considered to be a process of research to prove the possible physical onteraction between B2 receptor and dopamine D2 receptor in vivo, the results are interetong and signifficant.
Author Response
Reviewer #2 : ( Comments and Suggestions for Authors)
The authors revised the article as much as posible in adding the discussion by following the commnents of reviewers. Although this article is considered to be a process of research to prove the possible physical onteraction between B2 receptor and dopamine D2 receptor in vivo, the results are interetong and signifficant.
RESPONSE: We would like to express our sincere appreciation for your insightful comments and suggestions on our manuscript. Your guidance played a pivotal role in enhancing the quality of our article. We have carefully addressed each of your points, and your feedback greatly contributed to the development of a more comprehensive discussion. Once again, thank you for your valuable input and time dedicated to reviewing our work. We sincerely appreciate your expertise and constructive feedback.

Round 3
Reviewer 1 Report
Comments and Suggestions for Authors
The response to the authors improves discussion. It can be published.